# WAX INDUCER 1 Regulates β-Diketone Biosynthesis by Mediating Expression of the *Cer-cqu* Gene Cluster in Barley

**DOI:** 10.3390/ijms24076762

**Published:** 2023-04-05

**Authors:** Sophia V. Gerasimova, Ekaterina V. Kolosovskaya, Alexander V. Vikhorev, Anna M. Korotkova, Christian W. Hertig, Mikhail A. Genaev, Dmitry V. Domrachev, Sergey V. Morozov, Elena I. Chernyak, Nikolay A. Shmakov, Gennady V. Vasiliev, Alex V. Kochetov, Jochen Kumlehn, Elena K. Khlestkina

**Affiliations:** 1Institute of Cytology and Genetics, Siberian Branch of the Russian Academy of Sciences, 630090 Novosibirsk, Russia; 2Vavilov Institute of Plant Genetic Resources (VIR), 190000 Saint Petersburg, Russia; 3Leibniz Institute of Plant Genetics and Crop Plant Research (IPK), 06466 Gatersleben, Germany; 4N. N. Vorozhtsov Novosibirsk Institute of Organic Chemistry, Siberian Branch of the Russian Academy of Sciences, 630090 Novosibirsk, Russia

**Keywords:** CRISPR-associated endonculeases, cuticle, gene cluster, lipids, targeted mutagenesis, transcription factor, wax

## Abstract

Plant surface properties are crucial determinants of resilience to abiotic and biotic stresses. The outer layer of the plant cuticle consists of chemically diverse epicuticular waxes. The WAX INDUCER1/SHINE subfamily of APETALA2/ETHYLENE RESPONSIVE FACTORS regulates cuticle properties in plants. In this study, four barley genes homologous to the *Arabidopsis thaliana AtWIN1* gene were mutated using RNA-guided Cas9 endonuclease. Mutations in one of them, the *HvWIN1* gene, caused a recessive glossy sheath phenotype associated with β-diketone deficiency. A complementation test for *win1* knockout (KO) and *cer-x* mutants showed that *Cer-X* and *WIN1* are allelic variants of the same genomic locus. A comparison of the transcriptome from leaf sheaths of *win1* KO and wild-type plants revealed a specific and strong downregulation of a large gene cluster residing at the previously known *Cer-cqu* locus. Our findings allowed us to postulate that the WIN1 transcription factor in barley is a master mediator of the β-diketone biosynthesis pathway acting through developmental stage- and organ-specific transactivation of the *Cer-cqu* gene cluster.

## 1. Introduction

The surface of plants is covered by a cuticle that constitutes a protective layer composed of a cutin-based matrix and epicuticular waxes. The physiological properties of the cuticle are related to a variety of agronomic traits, including tolerance to drought and UV light, as well as resistance to pathogens [1]. The genetic control of cuticle wax accumulation in barley has been studied since the 1950s, when a number of wax-deficient mutants (*eceriferum*, *cer* mutants) were first obtained and characterized. According to complementation analysis, there are 79 different *cer* genomic loci in barley [2]. Despite many known *cer* mutants, only a few genes associated with these mutations have been identified thus far [3,4]. Among them is the *Cer-cqu* gene cluster known for leaf sheath wax accumulation control [5]. This cluster comprises three genes encoding enzymes of β-diketone biosynthesis. These are (1) the chalcone synthase-like PKSIII designated either as DKS (diketone synthase) or DMP (diketone metabolism-PKS) and located at the *Cer-c*/*Gsh6* locus, (2) a lipase/hydrolase designated as DMH (diketone metabolism-hydrolase) and located at the *Cer-q*/*Gsh1* locus, and (3) a cytochrome P450 enzyme with predicted hydroxylating activity designated as DMC (diketone metabolism-CYP450) and located at the *Cer-u/Gsh8* locus. A few other *cer* mutations have been mapped with high accuracy [3,6,7,8]; however, to date, the genetic regulatory mechanisms for barley wax accumulation have remained elusive.

The two major epicuticular wax compounds in barley are C-26 alcohol (hexacosanol) and C-31 β-diketone (hentriacontane-14,16-dione) [9]. The alcohol-forming biochemical pathway is homologous to the *Arabidopsis thaliana* wax synthesis machinery [10]. *A. thaliana* epicuticular wax is produced via fatty acid (FA) synthesis in plastids as well as FA elongation and further processing in the endoplasmic reticulum. The processing involves two major pathways leading to either alkanes, secondary alcohols and ketones or to primary alcohols and their esters [11]. The pathway leading to β-diketones does not exist in *A. thaliana*. It involves β-diketone synthase (DKS) along with related polyketide biochemical machinery and leads to the formation of β-diketones, hydroxy-β-diketones and alkan-2-ol esters [12]. Three enzymes of polyketide component biosynthesis are encoded by genes forming the aforementioned *Cer-cqu* gene cluster at chromosome 2H in barley [5,13].

The presence of a metabolic gene cluster for β-diketone biosynthesis in the barley genome suggests the coordinated trans-regulation of this cluster through specific transcription factors [14]. The AP2/ERF transcription factor responsible for the activation of wax deposition and cutin synthesis (named WIN1 for WAX INDUCER) was identified in *A. thaliana*. The known genes of wax biosynthesis enzymes were activated in WIN1-overexpressing plants [15]. An alternative name of the WIN1 transcription factor is SHINE (SHN), inspired by the shiny appearance of the mutant [16]. WIN1/SHN1-like AP2/ERF transcription factors are widely present in plants and known to be involved in cuticle organization and wax deposition [17,18]. *HvWIN1* and *HvNUD1* are the two annotated *WIN1/SHN1* genes in barley. The *NUD1* gene has been identified as a regulatory gene responsible for lipid accumulation at the surface of the pericarp, defining the naked/hulled grain phenotype [19,20]. The *WIN1* gene was shown to be related to Fusarium resistance and FA synthesis in spikes and wax biosynthesis regulation [21,22].

The *WIN1* gene and its closest paralogues are candidates for the regulation of cuticle wax accumulation in barley. In the present study, a gRNA/cas9 construct targeting conserved motifs in four AP2/ERF genes, including *WIN1* and its homologues, was used to generate mutant plants. The mutant glossy leaf sheath phenotype was associated with *win1* loss of function in segregating M2 plants. Isogenic mutant lines differing from the wild-type (WT) line only in the *WIN1* gene sequence were obtained in the M3 generation. The *win1* knockout (KO) mutants exhibited a glossy sheath phenotype typical of *cer* mutants, downregulation of the *Cer-cqu* gene cluster and a deficiency in product accumulation of the β-diketone biosynthesis pathway. In addition, a complementation test confirmed that the *WIN1* gene is allelic to the polymorphic *Cer-x* locus.

## 2. Results

### 2.1. Mutations in the WIN1 Gene Cause a Glossy Leaf Sheath Phenotype

An approach based on the simultaneous knockout of a few homologous genes was implemented in this study. A gRNA was designed to direct Cas9 nuclease to target motifs in four genes belonging to the AP2/ERF family (Figure 1a) including the previously known *NUD1* (HORVU7Hr1G089930, Taketa et al., 2008) and *WIN1* (HORVU6Hr1G038120, Kumar et al., 2016) genes and two genes with unknown functions (HORVU6Hr1G085850 and HORVU7Hr1G029870 loci). Different combinations of mutant alleles of the four target genes were obtained in M1 (T0) and M2 (T1). Two independently occurring mutant phenotypes were observed in the population of mutant plants: naked grain and glossy sheath, associated with visible cuticle wax deficiency. The naked grain phenotype was connected with mutations in the *NUD1* gene [20], and the glossy leaf sheath phenotype was invariably associated with mutations in the *WIN1* gene. All independently obtained *win1* mutants expressed the same phenotype. The wax deficiency is not seen during the vegetative phase of development. At the booting stage, the flag leaf sheath looks glaucous in the WT and bright green, glossy, in *win1* KO mutants. At the heading stage, mutant plants exhibit green glossy internodes and spikes without any signs of wax accumulation. Water drops remain at the surface of spike and leaf sheaths of the mutants (but not at the surface of their leaf blade), while WT surfaces have no water droplet retention phenotype. In contrast, mutations in the two unknown related genes did not cause any visible phenomena.

In the progeny of two out of 45 M2 plants harboring different mutations in the *WIN1* gene (plant identifiers 17-4 and 25-2), it was possible to select T-DNA-free M3 individuals carrying homozygous translational frameshift mutations in the *WIN1* gene along with wild-type sequences in the other three target genes. These individuals represent non-segregating, isogenic mutant lines (Figure 1b) and gave rise to progeny plants that were used for further analyses. The overall developmental progress of the KO mutant lines did not deviate from that of the WT line (Appendix A). Likewise, major agronomical characteristics measured in greenhouse-grown plants were not apparently affected by the loss of function of *WIN1* (Appendix A). The progeny of the M3 plant 25-2-2 carrying a homozygous, single-nucleotide deletion at the Cas9 cleavage site of the *WIN1* target motif was used to generate transcriptome data, as well as for a detailed analysis of the wax components.

### 2.2. The win1 KO-Associated Glossy Leaf Sheath Phenotype Is Recessive

Four F1 plants were obtained by crossing T-DNA-free, homozygous mutants (*win1*-KO Line 25-2-2, M4), and cv. “Golden Promise” WT. Phenotyping was performed based on visual inspection (Appendix A). All hybrids consistently exhibited a normal glaucous phenotype identical to the wild-type. Sanger sequencing of the Cas9 target motif showed that all hybrids were heterozygous, i.e., a wild-type allele was combined with the *win1* KO allele (1-bp deletion) derived from Line 25-2-2. A segregating F2 generation was obtained through self-pollination of these F1 hybrids. A total of 114 siblings were obtained, 85 of which were of the normal glaucous phenotype, and 29 showed cuticular wax deficiency with a glossy sheath phenotype. The phenotypic segregation in the F2 population was 3:1. The consistent phenotype of F1 hybrids and the proportional occurrence of the two phenotypes among the F2 siblings strongly suggest a monogenic recessive inheritance of the glossy leaf sheath phenotype.

### 2.3. WIN1 Is Allelic to Cer-x

The clear glossy sheath phenotype of the *win1* KO plants (Figure 1c) suggests that one of the known *Cer* loci does correspond to the *WIN1* gene. *WIN1* is located on barley chromosome 6H, while there is only one *Cer* locus associated with a similar phenotype on the same chromosome, namely *Cer-x* or *GSH4* (https://bgs.nordgen.org/index.php?pg=bgs_show&docid=10, accessed on 1 August 2022). To check whether *WIN1* and *Cer-x* are allelic to each other, *win1* KO mutants were crossed with *cer-x* mutants from the NordGen GenBank. Heterozygosity was confirmed using Sanger sequencing. All F1 plants exhibited the glossy sheath phenotype, strongly suggesting that the *WIN1* gene corresponds to the *Cer-x* or *GLOSSY SHEATH 4* gene (Figure 1d–i). The analysis of the *WIN1* gene structure in *cer-x* mutants revealed a T-to-A nucleotide substitution in comparison to cvs. Bonus and Bowman wild-type *WIN1*. This mutation led to an amino acid change from tryptophan to arginine (Figure 1j). These amino acids not only belong to different types but also reside in a highly conserved AP2 domain responsible for DNA binding and necessary for protein function as a transcription factor. The mutation is likely to prevent protein binding to DNA and cause the loss of *WIN1* function in the *cer-x* mutants.

### 2.4. win1 Mutants Are Deficient in Tubule-Shaped Microscopic Cuticle Structures

Scanning electron microscopy was performed for the leaf sheath, leaf blade and lemma abaxial and adaxial surfaces at the tillering stage and heading stage (Figure 2a and Appendix A). At the tillering stage, both leaf blades and sheaths were covered by platelet-shaped epicuticular structures. The same structures also covered leaf blades at the heading stage but not leaf sheaths, stems or spikes. In WT plants, the leaf sheaths and spikes at the heading stage are covered by cuticle structures that looked similar to tiny branching tubules. The mutant plants featured a reduced amount of such structures at the surface of the leaf sheath and lemma. According to these results, the *WIN1* gene was responsible for the accumulation of tubule-type wax structures during generative development of leaf sheaths and spikes.

### 2.5. win1 KO Plants Showed Reduced β-Diketone and Related Wax Compound Coverage

Since macroscopically and microscopically visible differences in epicuticular wax occur at the reproductive phase, we decided to investigate the wax properties of generative tillers. The analysis of leaf blade wax composition revealed the presence of the following compound classes in both the mutant and WT: alkanes, alkylresorcinols, aldehydes, primary alcohols and FAs (Figure 2b, Appendix A). In both the mutant and WT, the major wax compounds were of the very long-chain (VLC)-saturated C22–C30 alcohol class, with C26 hexacosanol being the most abundant. In mutants, the total amount of leaf blade wax proved to be reduced by 25% on average. However, all compound classes and all detectable individual compounds were present in similar proportions compared with the WT. Consequently, the difference between *win1* KO and WT leaf blade wax could be classified as a quantitative decrease without conspicuous qualitative effects.

In total, the *win1* KO mutant leaf sheath surface accumulated about sevenfold less wax than the WT. Detected compound classes included alkanes, alkylresorcinols, FAs, aldehydes, primary alcohols, alkan-2-ols and β-diketones (Figure 2b, Appendix A) with predominant accumulation of C31 14,16-hentriacontanedione (about 80% at average). In addition, C29 and C33 β-diketones were identified in trace amounts; apparently, these β-diketones were nonacosane-14,16-dione and tritriacontane-16,18-dione, respectively [23]. Fragments of nonacosane-14,16-dione, hentriacontane-14,16-dione and tritriacontane-16,18-dione mass spectra are shown in Appendix A. The C31 β-diketone proved most severely affected in the *win1* KO, with its amount being approximately 18-fold lower in mutants. Alkan-2-ol (derived from the alkan-2-ol esters) amounts were also drastically reduced in mutants. Further compounds found to be significantly lower in mutants were alkylresorcinols and aldehydes as well as C18 and C20 FAs and a few alkanes (Appendix A). The method of wax biochemical analysis used in this study quantifies FAs in the form of methyl esters as a sum of free FAs and FAs from alkan-1-ol esters and alkan-2-ol esters, and primary alcohols as a sum of free primary alcohols and hydrolysis products of primary alcohol esters. The detected change in alkan-2-ol content reflects a decrease in alkan-2-ol ester formation in mutants. Due to the strong alteration in the overall wax pattern, alcohols, instead of β-diketones, represented the major compound class in the *win1* KO mutants.

### 2.6. Transcriptome Analysis Reveals a High Number of WIN1-Dependent Genes

The transcriptome difference between WT and *win1* KO mutant plants was studied for leaf blade and leaf sheath central fragments. In total, the expression of 26,807 genes was higher than the significance threshold. The analysis of differentially expressed genes (DEGs) was conducted based on two comparison groups: leaf blade WT vs. leaf blade *win1* (lb_wt vs. lb_*win1*, respectively) and leaf sheath WT vs. leaf sheath win1 (ls_wt vs. ls_*win1*, respectively). In the *win1* KO samples, there were 808 upregulated and 765 downregulated genes in leaf blades and 605 upregulated and 81 downregulated genes in leaf sheaths compared with their WT counterparts (Appendix A, Figure 3a). Among these DEGs, 201 were upregulated in both tissues, and 28 were downregulated in both tissues of the mutant. Three genes were upregulated in mutant leaf blades but downregulated in mutant leaf sheaths (HORVU6Hr1G073300, HORVU2Hr1G002580, and HORVU2Hr1G002630), whereas three genes (HORVU6Hr1G002390, HORVU7Hr1G101310, and HORVU3Hr1G047220) were downregulated in leaf blades and upregulated in leaf sheaths of the mutant (Figure 3b). The expression of 16 genes was confirmed using qRT-PCR, including 7 genes of the *Cer-cqu* cluster and 9 randomly selected differentially expressed genes. The measured expression of these genes was highly consistent with the deep-sequencing data (Spearman’s correlation of 0.89) (Figure 3c and Appendix A).

### 2.7. A WIN1-Dependent Set of Enzyme-Encoding Genes Is Contained in a Five-Mb Region of Chromosome 2H

Since *win1* KO plants largely lack β-diketones, the expression of *Cer-cqu* cluster genes involved in β-diketone biosynthesis was of primary interest. All these genes exhibit a highly contrasting expression in the WT, being silent in the leaf blade and highly active in the leaf sheath. In the *win1* KO background, the *Cer-cqu* cluster genes are expressed at low level in both leaf blade and leaf sheath, being slightly more active in the latter (Figure 4, Appendix A).

Along with the three previously known genes of the *Cer-cqu* cluster, seven additional genes and two pseudogenes of the same chromosomal region exhibited a similar expression pattern (Figure 4; Appendix A). An MAH1-like cytochrome alkane hydroxylase (HORVU2Hr1G002580) gene and two *Cer-u*-homologous pseudogenes (HORVU2Hr1G002570, HORVU2Hr1G002610) reside upstream of the *Cer-cqu* cluster. One gene encoding a transmembrane receptor from the AdipoR/Haemolysin-III (HlyIII)-related family (HORVU2Hr1G002630) and one gene encoding a desmethyl-deoxy-podophyllotoxin synthase-like Cytochrome P450 superfamily protein (HORVU0Hr1G001800) were found in the 5′- and 3′-adjacent regions, respectively. Another two co-expressed genes were located 1 Mb downstream of the cluster; they encoded a CER-C-like polyketide synthase (HORVU2Hr1G003280) and a putative carboxylesterase related to CER-Q (HORVU2Hr1G003300). Another two co-expressed genes were found to be located more than 3.5 Mb downstream. They encoded a type III PKS (HORVU2Hr1G004690) homologous to a rye alkylresorcinol synthase [25] (Appendix A) and a O-methyltransferase (HORVU2Hr1G004710) equal to known barley acetylserotonin O-methyltransferase 1-like (XP_044963919.1). The molecular function of this enzyme in barley in unknown. Taking into account that the chemical structure of acetylserotonin is similar to resorcinol, it can be assumed that this enzyme may act as a alkylresorcinol O-methyltransferase. In total, twelve co-expressed genes and pseudogenes resided on a chromosomal region of approximately 5 Mb in length.

### 2.8. Barley Wax-Related Genes Exhibit Tissue-Specific Expression Patterns

To assess the impact of the WIN1 transcription factor on the entire wax biosynthesis machinery, the expression of barley genes putatively related to wax accumulation was analyzed with a particular focus on genes specifically expressed in the leaf sheath (Appendix A). The *Win1* gene itself did not show a significant difference in expression between either mutant and WT samples, or leaf sheath and leaf blade, which was confirmed also by qRT-PCR. Other AP2/ERF transcription factors of the WIN1/SHN1 subfamily also had low or no expression in examined tissues. Two MIXTA-like transcription factors shown to be related to WIN1/SHN1 function in *A. thaliana* [26] were found in the barley genome, and one of them is activated in the leaf sheath in both wild-type and mutant.

A search for orthologues in the barley genome identified counterparts for most *A. thaliana* genes known to be related to cuticle organization and wax biosynthesis (Figure 5; Appendix A). Wax biosynthesis in *A. thaliana* involves FA elongation and further processing. FA elongation includes four steps catalyzed by four main enzymes: 3-ketoacyl-CoA synthase (KCS), 3-ketoacyl-CoA reductase (KCR), β-hydroxyacyl-CoA dehydratase (HCD), and enoyl-CoA reductase (ECR) [11]. The largest barley gene family involved in this pathway encodes different forms of 3-ketoacyl-CoA synthase, the first acting enzyme in the FA elongation process [27]. The *WIN1* gene mutation differentially affected genes of this family (Figure 5a). There are two cloned and well-studied barley genes of KCS family, belonging to the KCS1 (HORVU4Hr1G063420) and KCS6 (HORVU4Hr1G067340) subfamilies [3,28]. Both of these genes did not show either tissue-specific or WIN1-dependent regulation. Two genes belonging to the KCS5 and KCS6 subfamilies (HORVU6Hr1G073300 and HORVU5Hr1G087530, respectively) proved to be leaf sheath-specific with a WIN1-related expression pattern similar to genes of the *Cer-cqu* cluster. The gene families corresponding to the next three steps of FA elongation are much less redundant in barley. Among these families, only one gene encoding KCR (HORVU7Hr1G024370) was specifically expressed in the leaf sheath.

After a certain number of FA elongation cycles, very long-chain fatty acids (VLCFA)-CoA are converted into final wax compounds [11]. Alkanes are formed from VLCFA-CoA by decarbonylation catalyzed by *CER1* and *CER3* in *A. thaliana*. One of the barley CER1 homologues was specifically expressed in the leaf sheath (HORVU1Hr1G039830), while CER3 gene expression did not show specific regulation (Figure 5b). The major compounds in barley leaf wax are very long-chain primary alcohols. Primary alcohols are synthesized via the action of fatty acyl-CoA reductase (FAR, *CER4*). The expression of seven barley homologues of *A. thaliana CER4* was checked; most of them were upregulated in the win1 KO leaf blade. Primary alcohols can further form esters via the action of wax ester synthase/diacylglycerol acyltransferase (WSD). Twelve genes homologous to *A. thaliana* WSD were found to be expressed in the analyzed tissues, and none of them proved WIN1-dependent. One gene from this family was strongly specific for the leaf sheath (HORVU7Hr1G024590), and one was specific for the leaf blade (HORVU3Hr1G074910). Midchain alkane hydroxylase 1 (MAH1), belonging to the cytochrome P450 family, is involved in the alkane-forming pathway of *A. thaliana* and catalyzes the formation of secondary alcohols and ketones [11]. One of the barley MAH1 gene homologues was located within the *Cer-cqu* cluster (HORVU2Hr1G002580), and another two showed an expression pattern similar to *Cer-cqu* cluster regulation (HORVU3Hr1G011830 and HORVU3Hr1G110180).

To reveal genes regulated by the WIN1 transcription factor that are possibly involved in wax phenotype formation, common upregulated DEGs were selected from three comparisons: ls_wt-ls_*win1*, ls_wt-lb_wt, and ls_wt-lb_*win1*. This group of genes exhibits expression specific for the leaf sheath and is downregulated in the *win1* KO mutant. In general, 38 genes with this type of regulation were found (Appendix A). Twelve of these do not belong to the *Cer-cqu* cluster and known wax-controlling genes and can be considered novel candidates for being WIN1-related. Most of them are poorly annotated; four encode enzymes, four are clustered together at chromosome 2H and encode wound-responsive family proteins, while the remaining ones are encoding proteins from various further families.

## 3. Discussion

### 3.1. WIN1/win1 Isogenic Lines as Experimental Model to Study Cuticle Wax Formation in Barley

*Eceriferum* (*cer*) mutants with different type of cuticle wax deficiency have been used as models to elucidate the genetic control of wax accumulation in barley. Different types of cuticle wax deficiency in *cer* mutants (reduction in wax at the spike and upper leaf sheaths, and either at leaf blades, spikes or the entire plant) suggests different regulatory mechanisms for wax accumulation at the spike, upper leaf sheaths of reproductive shoots and leaf blades [2]. The targeted mutagenesis of the *WIN1* gene led to a glossy sheath phenotype typical for *eceriferum* mutants in barley, and a complementation test revealed that the *WIN1* gene is allelic to the previously identified *Cer-x* locus. The *WIN1/win1* isogenic lines generated in the present study constitute a perfect experimental model, because all differences in phenotype or gene expression seen in these lines can be attributed to a single mutation. The *win1* KO lines feature a biochemical wax composition similar to that of *cer-x* mutants including a great reduction in β-diketone amounts [29] and a change in alkane composition (according to the Nordgen BGS 354 description). General growth performance is not affected in the *win1* KO lines in contrast to *cer-x* mutants, with the latter showing reduced grain yield and kernel weight (Nordgen BGS 354 description). The growth inhibition in the conventional *cer-x* mutants may be explained by adverse effects caused by their high load of non-specific mutations.

The leaf sheath and leaf blade of barley reproductive shoots produce different types of cuticle wax. The comparison of gene expression and cuticle wax component accumulation in these organs across the two isogenic lines allowed us to distinguish between genes involved in two major pathways of cuticle wax biosynthesis in barley and to thereby identify a large gene cluster controlling the β-diketone-related biochemical pathway.

### 3.2. The WIN1 Transcription Factor Specifically Regulates the β-Diketone Biosynthesis Pathway

The epicuticle wax biosynthesis machinery in barley includes two different biochemical systems functioning in different plant organs and developmental stages. The system producing alcohol-enriched platelet-shaped wax structures is homologous to the *A. thaliana* wax biochemical pathway [30]. In barley, this system operates during vegetative development as well as in leaf blades formed during the reproductive phase. Biochemical analysis of the leaf blade wax composition revealed a set of even-chained saturated FAs representing products of FA elongation, which takes place in the endoplasmic reticulum. The second class of substances constitutes long-chain fatty alcohols with the dominance of hexacosanol formed from the aforementioned class of FAs [31]. The amounts of chemical compounds in leaf blade wax of *win1* KO plants were only slightly lower in comparison to those in WT plants (Figure 2). This indicates that the WIN1 transcription factor is not essential for the regulation of wax biosynthesis in leaf blades. Intriguingly, the stem wax composition was different from that of leaf blades, although the same set of FA elongation products and related primary alcohols was present. The amounts of FAs with chain lengths from 22 to 34 and primary alcohols with chain lengths from 22 to 26 were similar in the stem wax of both the *win1* mutant and WT (Appendix A). It can be assumed that these compounds are also products of the alcohol-forming wax biosynthesis pathway. Most likely, this pathway is active in both the leaf blade and leaf sheath, and the WIN1 transcription factor has only a minor and probably indirect effect on this pathway. This conclusion is also supported by a comparison of the expression levels of genes known to participate in leaf wax biosynthesis. The key enzymes involved in FA elongation are members of the KCS family catalyzing the first step in every elongation cycle in a chain length-dependent manner. Two KCS barley genes are known to be involved in leaf wax biosynthesis. The *GLOSSY LEAF 1/ECERIFERUM-zh* gene (HORVU4Hr1G063420) encoding the HvKCS1 enzyme is specific for acyl chain lengths between 16 and 20 carbons [3], and the *HvKCS6* gene (HORVU4Hr1G067340) is responsible for VLCFA elongation from C24 to C26 and corresponding C26 alcohol formation [28]. These genes are expressed in both tissues and genotypes (Figure 5). Two other KCS genes from KCS4 and KCS10 subfamilies (HORVU1Hr1G089710 and HORVU4Hr1G076940, respectively) were also actively expressed in all samples. Among other genes of the FA elongation complex, it was also possible to detect some with expression in all examined tissues (HORVU6Hr1G054000 for KCR, HORVU3Hr1G116200 for HCD, and HORVU1Hr1G013970 for ECR). These genes can be considered as being part of the genetic mechanism controlling FA elongation and fatty alcohol biosynthesis.

The alkane-forming biochemical pathway in barley has only a minor contribution to cuticle wax composition. By contrast, this pathway plays a major role in *A. thaliana* cuticle wax formation and is activated by the WIN1/SHN1 transcription factor [16]. The alkane amounts were found to be reduced in the *win1* KO barley plants, suggesting an activating role of the WIN1 transcription factor for alkane biosynthesis.

The main phenotypic difference between WT and *win1* KO plants is the lack of β-diketones and related compounds at the surface of *win1* KO leaf sheaths. The biosynthesis of β-diketones in barley is implemented through a mechanism that is distinct from the alcohol-forming and alkane-forming pathways. This mechanism is not yet fully understood. Hentriaconane-14,16-dione is usually a major product of the β-diketone synthesis pathway. The other products are the hydroxylated forms of this β-diketone (25-hydroxy-hentriacontane-14,16-dione), alkan-2-ol esters, and minor C29 and C33 diketones. All aforementioned compounds were greatly reduced in *win1* KO leaf sheaths (Figure 2 and Appendix A). Taken together, the data strongly suggests the stage- and organ-specific regulation of the entire β-diketone pathway by the *WIN1* gene.

### 3.3. The Extended Cer-cqu Cluster Comprises Twelve Genes and Pseudogenes Regulated by the WIN1 Transcription Factor

The uniform regulation of a co-localized set of enzyme-encoding genes allows for the conclusion that the *Cer-cqu* cluster accounts for a larger chromosomal region than previously reported and includes a total of ten genes and two pseudogenes regulated by the WIN1 transcription factor in a tissue-specific manner (Figure 4, Appendix A). The presence of the functional *WIN1* gene causes a high-level of cluster activation in leaf sheaths. Nine of the functional genes encode enzymes, and one encodes a transmembrane receptor (HlyIII) possibly related to the regulation of lipid metabolism [32]. The *Cer-c*, *Cer-q* and *Cer-u* genes represent the core of the cluster and are responsible for the main reactions in the β-diketone biosynthesis pathway. Genes from the cytochrome P450 superfamily are located around the core cluster. Another two pairs of co-expressed genes, including *Cer-c* and *Cer-q* homologues, reside at different distances downstream of the core, thereby forming a peripheral zone of the cluster. These results are perfectly consistent with the previous findings of Hen-Avivi et al.’s (2016) work, where the wheat *W1* locus and the corresponding barley chromosome 2H region were compared and wide multigenic diketone synthesis clusters were predicted in both species [13]. The functions of the genes residing in the peripheral zone of the cluster remain to be elucidated. A particularly interesting observation made in this study is the presence of a gene related to alkylresorcinol biosynthesis within the distal part of the WIN1-activated *Cer-cqu* cluster, and the reduction in alkylresorcinol contents in the *win1* KO leaf sheaths. The possible association between β-diketone and alkylresorcinol biosynthesis was discussed previously [33], but no experimental evidence supporting this association was available. The predominant accumulation of C17 alkylresorcinol in barley leaf sheath epicuticular wax was found in this study and was also observed in rye [25]. Our findings suggest that the cluster contains genes controlling different synthesis pathways based upon the activity of polyketide synthases. It can be assumed that specific alkylresorcinol biosynthesis pathway associated with diketone accumulation is active in leaf sheath.

### 3.4. The WIN1 Transcription Factor Is Involved in Complex Regulatory Mechanisms

The mechanism of WIN1-mediated transactivation of the extended *Cer-cqu* cluster remains elusive. The mRNA level of the *WIN1* gene was similar across all analyzed samples. This observation indicates the absence of self-regulation at the transcriptional level and the possible presence of an organ- and stage-specific partner (repressor or activator) cooperating with WIN1 at the posttranscriptional level. Mutation in the *WIN1* gene led to strong suppression of the *Cer-cqu* cluster expression in leaf sheath and simultaneous moderate activation in leaf blade (Figure 4). It indicates the role of the WIN1 transcription factor in appropriate tissue-specific regulation of wax biosynthesis, and opposite effect on the *Cer-cqu* cluster expression depending on organ and possibly developmental stage (Figure 6). It is difficult to predict the molecular mechanism of the WIN1 function. The published experimental data concerning the possible regulation of plant gene clusters are very poor. Co-expression of clustered genes encoding enzymes that act in a certain biochemical pathway has been shown for many plant species. However, only a few transcription factors regulating gene clusters have been identified in plants thus far. It usually remains unclear whether these factors act directly or have intermediates [14]. One known example of the co-regulation of clustered genes by different transcription factors is the synergistic activation of GAME9 AP2/ERF and SlMYC2 factors on the promoters of a few glycoalkaloid-related genes [34]. The coordinated action of the WIN1 and MIXTA-like transcription factors was shown in *A. thaliana* [26]. In the current study, the expression of one MIXTA-like transcription factor (HORVU6Hr1G060650) was found to be up-regulated the WT leaf sheath and further tissue-specific increase in its expression was observed in the *win1* mutant (Appendix A). Such expression pattern may indicate a link of the MIXTA-like factor to the WIN1 gene function as a co-activator of the β-diketone synthesis pathway in the leaf sheath.

## 4. Materials and Methods

### 4.1. Plant Material

The *win1* KO mutants were obtained using the two-rowed spring-type, British barley (*Hordeum vulgare* L.) cv. “Golden Promise”. For a complementation test, 60 *cer-x* mutants with the genetic backgrounds of cvs. “Bonus” (NGB 110944) and “Bowman” (Bowman near isogenic line 126, NGB 20532) from the NordGen GenBank (Center for Genetic Resources of the Nordic Countries) were used. Plant material was grown in the greenhouse complex of the ICG SB RAS (Center for Collective Use “Laboratory of Artificial Plant Cultivation”) with 12 h of illumination per day at 20–25 °C.

### 4.2. Generation of Homozygous win1 KO Lines

The motif (ggagacccaggagccccagTGG) identical for four barley genes from AP2/ERF family (loci HORVU6Hr1G038120, HORVU6Hr1G085850, HORVU7Hr1G029870 and HORVU7Hr1G089930) was selected as target for RNA-guided Cas9 endonuclease-mediated mutagenesis. The double-stranded oligonucleotide (5′-GGAGACCCAGGAGCCCCAG-3′) with 5′-overhangs was cloned into pSH121 generic vector between two BsaI restriction sites, which resulted in the vector NUD-45RGEN [35]. The final construct for *Agrobacterium*-mediated transformation was created by SfiI-cleavage of the p6i-2x35s-TE9 binary (DNA Cloning Service, Hamburg, Germany) and NUD-45RGEN vectors and ligation of fragments corresponding to vector backbone and cas9/gRNA expression cassette. *Agrobacterium*-mediated transformation was performed according to the work of Marthe et al. [36], through which 46 M1 (T0) plants with confirmed transgene insertion were obtained [20]. Eight of the M1 plants were selected for further propagation. Five grains from each of these primary mutant plants were used to raise M2 plants in the greenhouse. The glossy leaf sheath phenotype was observed in progenies derived from seven out of the eight primary mutants. Sanger sequencing revealed a perfect co-presence of the glossy leaf sheath phenotype and mutations in the *WIN1* gene, whereas no such correlation was seen with mutations in the other target loci. Two independent M2 plants (17-4 and 25-2) with a glossy leaf sheath phenotype were selected for further segregation of the *win1* KO mutation. These two plants were T-DNA-free according to PCRs for gRNA and cas9 and exhibited hulled grains (primers used for sequencing and transgene detection are listed in Appendix A). Thirty-nine and twenty-five M3 plants were obtained from these two individuals. All of these M3 plants were analyzed using Sanger sequencing for each target locus, which identified three T-DNA-free M3 lines per selected M2 parent (17-4-14, 17-4-19, and 17-4-21, and 25-2-2, 25-2-13, and 25-2-18) with homozygous mutations in the WIN1 gene and no mutation in the other related target genes.

### 4.3. General Phenotypic Analysis of the win1 KO Lines

M4 generation of the mutant lines 17-4-14, 25-2-2 and 25-2-18 and the isogenic “Golden Promise” WT control line were used for phenotypic analysis. The following phenotypic features were analyzed: total number of tillers and number of fertile tillers per plant, height of main tiller (cm, from the soil surface to the top of the spike, not including awns), length of main spike (cm, from uppermost node, not including awns), spikelet number of main spike, grain number of main spike (GNS), grain weight of main spike (GWS), 1000-grain weight (calculated as 1000-fold GWS divided by GNS), grain number per plant, and grain yield per plant. Additionally, the following phenological growth stages [37] were determined visually for each plant (days after sowing): cracking date (09 BBCH), tillering date (23 BBCH), booting date (47 BBCH), heading date (51 BBCH), milk date (75 BBCH), dough date (85 BBCH), and overripe date (97 BBCH). Samples of 10 plants were analyzed for the control line and for each of the *win1* KO lines. Means and standard deviations were calculated for each phenotypic trait and each growth stage. The significance of the differences between the *win1* KO lines and the control line was evaluated using the parameter-free Mann–Whitney U-test.

### 4.4. Analysis of Mutant Phenotype Inheritance and Complementation Test

For glossy sheath phenotype inheritance analysis, the homozygous mutant plants (*win1* KO Line 25-2-2, M4) and cv. “Golden Promise” WT plants were crossed, and the F1 hybrids and F2 plants obtained via self-pollination of F1 hybrids were analyzed through visual inspection. The genotype of hybrids was confirmed with Sanger sequencing.

For the complementation test, the following crosses were carried out: *win1* KO line 17-4-4 × NGB20532, *win1* KO line 25-2-2 × NGB20532, and *win1* KO line 25-2-2 × NGB110944. To analyze the obtained hybrids, phenotyping based on visual inspection and sequencing of the Cas9-addressed target motif of *WIN1* through Sanger sequencing was performed (primers used for sequencing are listed in Appendix A).

### 4.5. Scanning Electron Microscopy

Whole tillers of a *win1* KO plant (Lines 25-2-2, M4) and a control plant (“Golden Promise”, Lines 22-1) were cut at the tillering (BBCH 35) and heading (BBCH 45) stages for scanning electron microscopy (SEM). After cutting, intact tillers were air-dried at 40 °C for 24 h, and then fragments of leaf blades, leaf sheaths, and spikes (for heading stage) were sampled separately and stored in sealed Petri dishes. For SEM analyses, 3 × 3-mm fragments of lemmas, leaf sheaths, and leaf blades were prepared and placed at the sample holder under consideration of whether the abaxial or adaxial side was facing upwards. After coating with gold particles (Mini SC 7620, Quorum Technologies, Lewes, UK) according to the manufacturer’s instructions, samples were examined under a Carl Zeiss EVO MA 10 scanning electron microscope using SmartSEM software (Carl Zeiss, Oberkochen, Germany).

### 4.6. Wax Analysis

For biochemical analyses, the upper parts of generative tillers (BBCH code 45) from the second upper node to top were sampled: the spike was cut-off, two upper leaf blades were collected for leaf blade wax analysis, and the resting part of the tiller, including two upper leaf sheaths with culm, was collected for stem wax analysis. In total, four samples were collected for each experimental group. The 25-2-2 *win1* KO line was chosen for leaf blade wax analysis. The amount of plant material from one line was not sufficient to collect four samples for stem wax analysis. Therefore, materials from four *win1* KO lines (25-2-2, 25-2-13, 25-2-18 and 17-4-19) were used as four replicates for stem wax analysis. For wax extraction, leaf blades or stems were dipped in chloroform for 90 s. The chloroform extracts were dried using anhydrous sodium sulfate, filtered and evaporated in a rotary evaporator. Cuticular waxes obtained from the extraction were derivatized with 1% H_2_SO_4_ in MeOH at 80 °C for 3 h [38,39,40] and then analyzed using gas chromatography–mass spectrometry (GC-MS). The Mann–Whitney U test was used to determine significant differences in wax component contents. The details of analysis are presented in Appendix A [39,41,42,43,44,45,46,47,48,49].

### 4.7. RNA-seq Analysis

For gene expression analysis, 1 cm central fragments of flag leaf blade and flag leaf sheath were used. The plant material was collected from mutant line 25-2-2 (M4) and the control line at the booting stage (BBCH code 45). Four groups of experimental samples were formed: WT leaf blade (lb_wt), WT leaf sheath (ls_wt), *win1* KO leaf blade (lb_win1) and *win1* KO leaf sheath (ls_win1). The experiment was performed using four biological replicates, each containing tissue fragments pooled from three plants, with one fragment per plant. RNA was extracted from barley samples using the RNeasy Plant Mini Kit (QIAGEN, 74904) according to the manufacturer’s protocol. After DNAse I treatment, RNA-seq library preparations were carried out with 1 µg of total RNA using the TruSeq^®^ Stranded mRNA LT Sample Prep Kit (Illumina, San Diego, CA, USA) according to the manufacturer’s instructions. The quantity and quality of the libraries were assessed using an Agilent Bioanalyzer 2100 System and DNA High sensitivity kit (Agilent Technologies, Santa Clara, CA, USA). After normalization, the barcoded libraries were pooled and sequenced using a NextSeq550 sequencer and NextSeq^®^ 550 High Output v2 Kit with 75 cycles (Illumina). The sequence data processing is described in Appendix A [50,51,52,53,54,55,56], and RNA-seq data are presented in Appendix A.

## Figures and Tables

**Figure 1 ijms-24-06762-f001:**
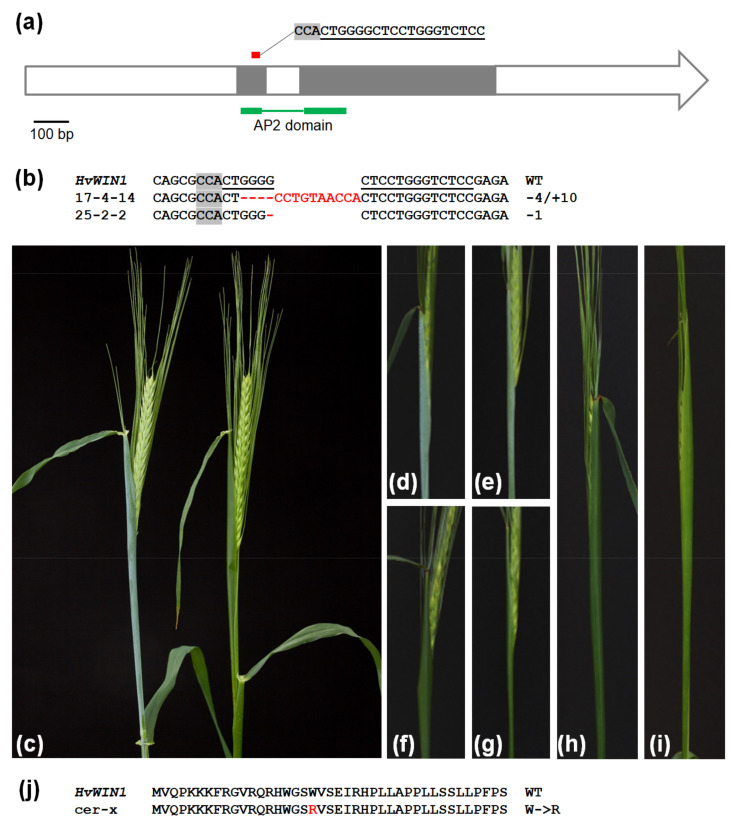
Genotype and phenotype of *win1* KO and cer-x.60 mutants. (**a**) Position and structure of the target motif in the *WIN1* gene. The coding sequence is shown in grey, AP2 domain in green, target motif in red. (**b**) *WIN1* gene mutations in the selected homozygous lines (Lines 17-4-14 and 25-2-2) are given as examples. Target motif is underlined and in bold; protospacer-adjacent motif is indicated by grey background. (**c**) “Golden Promise” wild-type (left) and glossy leaf sheath *win1* KO plants (right). (**d**) “Bonus” wild-type. (**e**) “Bowman” wild-type. (**f**) *cer-x.60* mutant in “Bonus”. (**g**) *cer-x.60* mutant in “Bowman”. (**h**) F1 *win1* KO × *cer-x.60* (in “Bonus”) hybrid. (**i**) F1 *win1* KO × *cer-x.60* (in “Bowman”) hybrid. (**j**) Fragment of the WIN1 protein structure in the wild-type (WT) and *cer-x.60* mutants.

**Figure 2 ijms-24-06762-f002:**
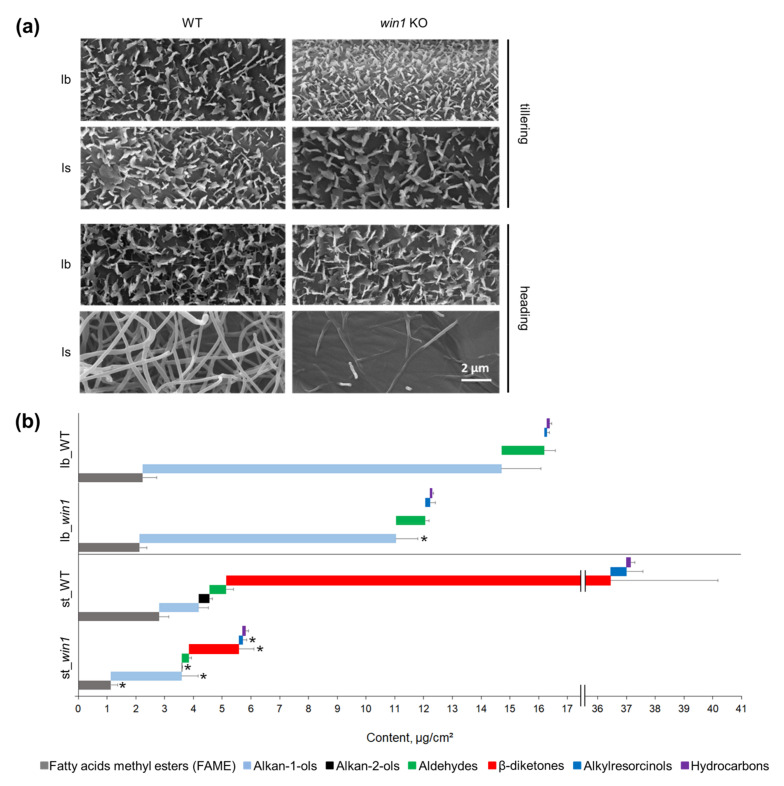
Cuticle wax analysis. (**a**) Scanning electron microscopy images of cuticular wax at different stages and organs. lb—leaf blade, adaxial surface, ls—leaf sheath, abaxial surface. (**b**) Chemical composition of cuticle wax in reproductive shoots of wild-type (WT) and *win1* knockout (KO) plants after derivatization. Average amounts of compound classes ±SD are presented. Lb—leaf blade, st—stem, asterisks indicate significant difference (*p* < 0.05) between *win1* KO and WT.

**Figure 3 ijms-24-06762-f003:**
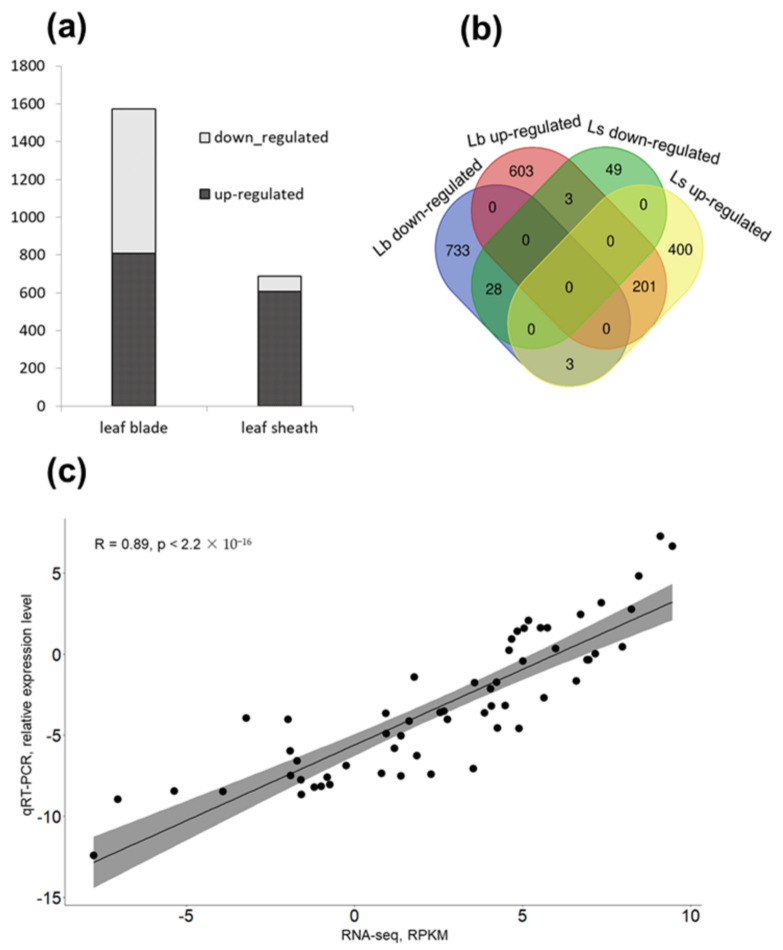
Analysis of differentially expressed genes (DEG). (**a**) Number of DEGs in pairwise comparison of WT vs. *win1* KO. (**b**) Venn diagram of DEGs in WT vs. *win1* KO comparison. (**c**) Correlation of qRT-PCR and RNA-seq data on gene expression. lb—leaf blade, ls—leaf sheath. RPKM—reads per kilobase of transcript, per million mapped reads.

**Figure 4 ijms-24-06762-f004:**
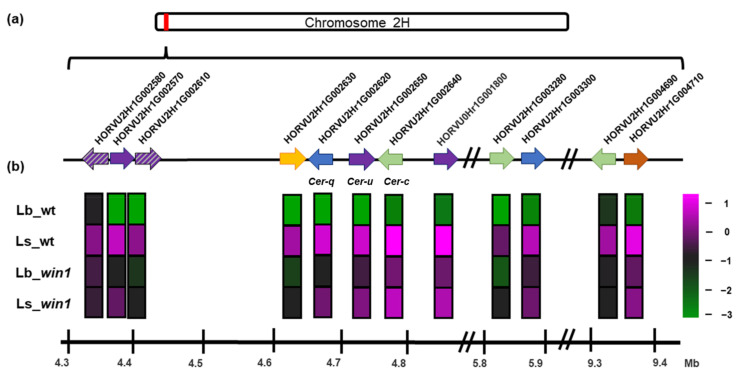
*Cer-cqu* gene cluster and co-expressed genes. (**a**) Physical map of the relevant region (highlighted in red colour) on barley chromosome 2H according to the Morex v2 assembly [24] Genes related to wax biosynthesis are coloured purple (cytochrome P450), blue (hydrolase), and orange (type-III polyketide synthase). The AdipoR/haemolysin-III-related coding gene is shown in yellow, the O-methyltransferase gene is shown in brown, pseudogenes are shown in striped patterns of corresponding color, and all other genes are shown in black. (**b**) Heatmap of gene expression. The magenta color indicates highly expressed genes, while green corresponds to low expression levels. The magenta-green gradient represents the change in fragments per kilobase of transcript per million mapped read (FPKM)-normalized log2-transformed counts from high to low. lb_*win1*—*win1* KO leaf blade; lb_wt—wild-type leaf blade; ls_*win1*—*win1* KO leaf sheath; and ls_wt—wild-type leaf sheath.

**Figure 5 ijms-24-06762-f005:**
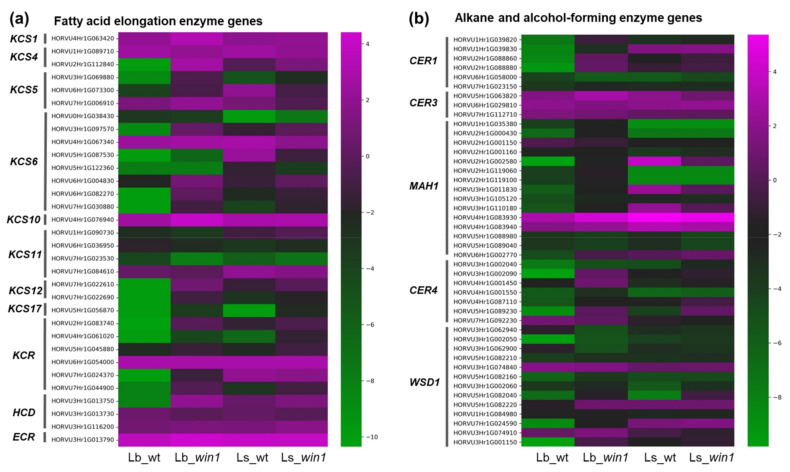
Expression heatmap of wax biosynthesis genes in wild-type and *win1* KO leaf blades and leaf sheaths. (**a**) Expression of genes involved in fatty acid elongation. KCS—3-ketoac-CoA synthase, KCR—3-ketoacyl-CoA reductase, HCD—b-hydroxyacyl-CoA dehydratase, and ECR—enoyl-CoA reductase. (**b**) Expression of alkane and alcohol-forming enzyme genes. CER1, CER3—acyl-CoA decarbonylases, MAH1—midchain alkane hydroxylase, CER4—fatty acyl-CoA reductase, and WSD1—wax ester synthase. The magenta color denotes highly expressed genes, and green indicates genes with lower expression. The magenta-green gradient represents the change in fragments per kilobase of transcript per million mapped reads (FPKM)-normalized log2-transformed counts from high to low. lb_*win1*—*win1* KO leaf blade; lb_wt—wild-type leaf blade; ls_*win1*—*win1* KO leaf sheath; and ls_wt—wild-type leaf sheath.

**Figure 6 ijms-24-06762-f006:**
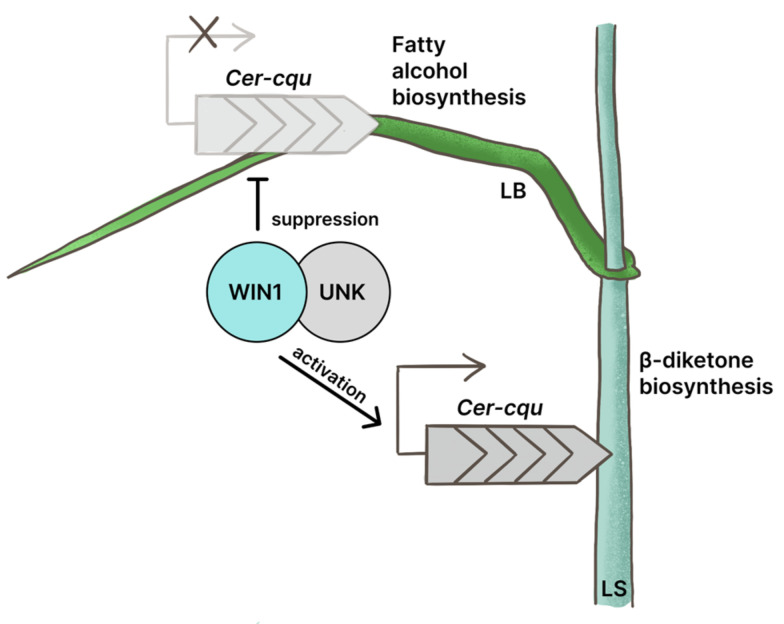
Involvement of WIN1 in the transcriptional regulation of the *Cer-cqu* gene cluster. LS—leaf sheath, LB—leaf blade, UNK—putative unknown partner.

## Data Availability

All relevant data can be found within the manuscript and its Appendix A.

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
