# Peer review of "WAX INDUCER 1 Regulates β-Diketone Biosynthesis by Mediating Expression of the Cer-cqu Gene Cluster in Barley"

_ijms, 2023, doi:10.3390/ijms24076762_

Round 1

Reviewer 1 Report

Lines 37-38 A reader is guided to conclude that while Cer-zh gene is in charge for wax biosynthesis, three Cer-cqu genes are in charge for wax accumulation. In fact, these are also biosynthesis genes as explained later in the text (lines 218-225). The authors may consider explaining the function of previously known Cer-cgu genes in more details in Introduction, instead of splitting the infromation between Introduction and Results section.

Line 207 - 'the expression of 16 genes was confirmed using qRT-PCR'. Were these gene selected arbitrary?

Figure 4. shows that in win KO background the expression of Cer-cqu gene cluster and co-expressed genes is upregulated in leaf blades and at the same time downregulated in leaf sheath. May you propose the explanation for the tissue specific role of WIN1 as regulator of these genes?

Lines 279-281 The description of up/downregulation of KCS genes in WIN1 mutants does not correspond to data in Fig. 5a?

Line 433 The 3.4. title is not adeuqate (repeated 3.3)

Minor technical comment - please note the placement od figure captions in supplement file - e.g. Figure FS5 caption is placed right below Figure FS4 (the right caption is in the previous page)

Author Response

Reviewer 1

  1. Lines 37-38A reader is guided to conclude that while Cer-zh gene is in charge for wax biosynthesis, three Cer-cqu genes are in charge for wax accumulation. In fact, these are also biosynthesis genes as explained later in the text (lines 218-225). The authors may consider explaining the function of previously known Cer-cgu genes in more details in Introduction, instead of splitting the infromation between Introduction and Results section.

Response: Corrected: Information on the function of genes of the Cer-cqu cluster is now given in the introduction at the end of its 1st paragraph. Additional references are added .for recent research papers, changes are highlighted in yellow.

  1. Line 207 - 'the expression of 16 genes was confirmed using qRT-PCR'. Were these gene selected arbitrary? –

Response: Corrected: The expression of 16 genes was confirmed using qRT-PCR, including 7 genes of the Cer-cqu cluster and 9 randomly selected differentially expressed genes (chapter 2.6, paragraph 1, highlighted in yellow).

  1. Figure 4. shows that in win KO background the expression of Cer-cqu gene cluster and co-expressed genes is upregulatedin leaf blades and at the same time downregulatedin leaf sheath. May you propose the explanation for the tissue specific role of WIN1 as regulator of these genes?

Response: Our findings do not allow to suggest a particular mechanism of WIN1 action. The effect of Win1 on tissue-specific Cer-cqu cluster activity and wax synthesis is discussed in the chapter “3.4. The WIN1 transcription factor is involved in complex regulatory mechanisms”. A schematic of WIN1 action is added as the new Fig. 6 (changes are highlighted in yellow).

  1. Lines 279-281 The description of up/downregulation of KCS genes in WIN1 mutants does not correspond to data in Fig. 5a?

Response: Corrected: Our erroneous statement is removed (chapter 2.8, paragraph 2).

  1. Line 433 The 3.4. title is not adeuqate (repeated 3.3)

Response: Corrected to “3.4. The WIN1 transcription factor is involved in complex regulatory mechanisms”

  1. Minor technical comment - please note the placement od figure captions in supplement file - e.g. Figure FS5 caption is placed right below Figure FS4 (the right caption is in the previous page)

Response: Corrected

Reviewer 2 Report

The manuscript investigated a group of four WIN genes potentially involved in wax biosynthesis in barley, using CRISPR-Cas9 and measurement of leaf wax composition and transcriptomics. The results are interestly and data is generally supporting the main conclusion. I have a few comments which I would bring out for the authors to consider:

1) Barley is a grass, using Arabidopsis thaliana as a reference might not be the best model, neither the best for gene homology comparison. Is there no similar work done in other monocot species, such as Rice or Wheat? On genome level as well as gene copy numbers, this could make quite a difference.

2) Not as an expert in plant cell wall research, but I would like to see more information on the biochemical side of the leaf sheath or leaf blade wax diketone biosynthesis. Knowing WIN1 as a ERF2 TF, one might anticipate to see some connection betweem WIN1 and downstream target genes in a model highlighting the connection to wax formation. Could bring a model at the end.

3)"glossy leaf sheath" as it is described in the text, is not well presented in Figure 1. I don't know if one could bring a higher resolution pic, or use some parameters to support this point. perhaps go for a reference with mutant of similar phenotype.

4) genetically, although the author claims WIN1 is allelic to previously identified Cer-x., they are different types of mutations. one as amino acid substitution, the other as frameshift? on protein level they are not equivalent, how does that correlate with its function as a TF protein?

5) Latin names, such as Arabidopsis thaliana; and mutant names, and genes, should be in italic font.

Author Response

Comments and Suggestions for Authors

The manuscript investigated a group of four WIN genes potentially involved in wax biosynthesis in barley, using CRISPR-Cas9 and measurement of leaf wax composition and transcriptomics. The results are interestly and data is generally supporting the main conclusion. I have a few comments which I would bring out for the authors to consider:

1) Barley is a grass, using Arabidopsis thaliana as a reference might not be the best model, neither the best for gene homology comparison. Is there no similar work done in other monocot species, such as Rice or Wheat? On genome level as well as gene copy numbers, this could make quite a difference.

Response: There are various nomenclatures of KCS family genes in different species including barley. For example, Tong et al. (2021) did not use gene numbers at all, but only relied on database accession numbers. Li et al. (2018) proposed their own system of barley KCS gene numbering, where the known genes KCS1 and KCS6 do correspond to A. thaliana numbering, wheras other genes were randomly given their numbers. To our opinion, the A. thaliana nomenclature is the most appropriate for the time being, because the first discovered barley KCS genes were named according to their A. thaliana orthologues. We prefer not to create a new nomenclature of barley genes at this point to avoid unnecessary complications and misunderstandings.

Tong, T.; Fang, Y. xia; Zhang, Z.; Zheng, J.; Zhang, X.; Li, J.; Niu, C.; Xue, D.; Zhang, X. Genome-Wide Identification and Expression Pattern Analysis of the KCS Gene Family in Barley. Plant Growth Regul. 2020, doi:10.1007/s10725-020-00668-3.

Li, C.; Haslam, T.M.; Krger, A.; Schneider, L.M.; Mishina, K.; Samuels, L.; Yang, H.; Kunst, L.; Schaffrath, U.; Nawrath, C.; et al. The β-Ketoacyl-CoA Synthase Hv KCS1, Encoded by Cer-Zh, Plays a Key Role in Synthesis of Barley Leaf Wax and Germination of Barley Powdery Mildew. Plant Cell Physiol. 2018, 59, 806–822, doi:10.1093/pcp/pcy020.

2) Not as an expert in plant cell wall research, but I would like to see more information on the biochemical side of the leaf sheath or leaf blade wax diketone biosynthesis. Knowing WIN1 as a ERF2 TF, one might anticipate to see some connection betweem WIN1 and downstream target genes in a model highlighting the connection to wax formation. Could bring a model at the end.

Response: Our findings established only the fact of WIN1-dependent Cer-cqu cluster regulation. The win1 mutation also affects expression of other putative wax biosynthesis genes (Figure 5), but since the functions of these genes are still unclear, no comprehensive model can be suggested so far. The effect of Win1 on tissue-specific Cer-cqu cluster activity and wax synthesis is discussed in the chapter “3.4. The WIN1 transcription factor is involved in complex regulatory mechanisms”. A schematic of WIN1 action is added as the new Fig. 6 (changes are highlighted in yellow).

3)"glossy leaf sheath" as it is described in the text, is not well presented in Figure 1. I don't know if one could bring a higher resolution pic, or use some parameters to support this point. perhaps go for a reference with mutant of similar phenotype.

Response: Done: A link to the cer-x mutant phenotype is given in chapter 2.3 (highlighted in yellow)..

4) genetically, although the author claims WIN1 is allelic to previously identified Cer-x., they are different types of mutations. one as amino acid substitution, the other as frameshift? on protein level they are not equivalent, how does that correlate with its function as a TF protein?

Response: Done: An explanation of the loss-of-function effect of the example of amino acid substitution is given in the chapter 2.3 (highlighted in yellow).

5) Latin names, such as Arabidopsis thaliana; and mutant names, and genes, should be in italic font.

Response: Corrected

Round 2

Reviewer 1 Report

The authors have adequately address all comments raised by this reviewer.